# The Polyubiquitin Gene *MrUBI4* Is Required for Conidiation, Conidial Germination, and Stress Tolerance in the Filamentous Fungus *Metarhizium robertsii*

**DOI:** 10.3390/genes10060412

**Published:** 2019-05-29

**Authors:** Zhangxun Wang, Hong Zhu, Yuran Cheng, Yuanyuan Jiang, Yuandong Li, Bo Huang

**Affiliations:** 1School of Plant Protection, Anhui Agricultural University, Hefei 230036, China; luckywang2011@ahau.edu.cn (Z.W.); chengyuran1993@163.com (Y.C.); zwbljiang1994@163.com (Y.J.); funliyuandong@163.com (Y.L.); 2Anhui Provincial Key Laboratory of Microbial Pest Control, Anhui Agricultural University, Hefei 230036, China; zhuhong@ahau.edu.cn

**Keywords:** *Metarhizium robertsii*, polyubiquitin, conidiation, conidial germination, stress tolerance

## Abstract

The polyubiquitin gene is a highly conserved open reading frame that encodes different numbers of tandem ubiquitin repeats from different species, which play important roles in different biological processes. *Metarhizium robertsii* is a fungal entomopathogen that is widely applied in the biological control of pest insects. However, it is unclear whether the polyubiquitin gene is required for fungal development, stress tolerance, and virulence in the entomopathogenic fungus. In the present study, the polyubiquitin gene (*MrUBI4*, MAA_02160) was functionally characterized via gene deletion in *M. robertsii.* Compared to the control strains, the *MrUBI4* deletion mutant showed delayed conidial germination and significantly decreased conidial yields (39% of the wild-type 14 days post-incubation). Correspondingly, the transcript levels of several genes from the central regulatory pathways associated with conidiation, including *brlA*, *abaA,* and *wetA*, were significantly downregulated, which indicated that *MrUBI4* played an important role in asexual sporulation. Deletion of *MrUBI4* especially resulted in increased sensitivity to ultraviolet (UV) and heat-shock stress based on conidial germination analysis between mutant and control strains. The significant increase in sensitivity to heat-shock was accompanied with reduced transcript levels of genes related to heat-shock protein (*hsp*), trehalose, and mannitol accumulation (*tps*, *tpp*, *nth,* and *mpd*) in the *MrUBI4* deletion mutant. Deletion of *MrUBI4* has no effect on fungal virulence. Altogether, *MrUBI4* is involved in the regulation of conidiation, conidial germination, UV stress, and heat-shock response in *M. robertsii*.

## 1. Introduction

Protein ubiquitination is a well-known process responsible for protein degradation and quality control, which regulates different biological pathways via the ubiquitin-proteasome system (UPS) [1,2]. The UPS is the primary cytosolic proteolytic machinery that selectively degrades various forms of misfolded or damaged proteins [3,4]. Of these, monomeric ubiquitin is a protein with 76 amino acids that is highly conserved in lower and higher eukaryotes [5]. Currently, ubiquitin-coding genes in yeast are divided into 2 major classes: polyubiquitin genes and monomeric ubiquitin fusion genes [6,7]. Polyubiquitin genes are arranged in tandem with head-to-tail repeats of 228 bp, and the number of tandem ubiquitin repeats varies between and within different species [8]. These polyubiquitin genes encode a polyubiquitin precursor protein, which is rapidly cleaved to produce mature ubiquitin monomers via deubiquitinating enzymes (DUBs) [9,10]. Conversely, the monomeric ubiquitin fusion gene only encodes a single ubiquitin repeat that is finally fused to unrelated peptide sequences [8]. Previous research in *Saccharomyces cerevisiae* suggested that the polyubiquitin gene (*UBI4*) encoded 5 tandem ubiquitin repeats and was highly induced under stress conditions, whereas the monomeric ubiquitin fusion genes were the primary contributors to cellular ubiquitin synthesis under normal conditions [6,11].

Recently, the polyubiquitin gene was shown to play an important role in fungal development, stress resistance, and virulence [12]. Previously, polyubiquitin gene expression at the transcription level was observed to be significantly upregulated under environmental stress in different pathogenic fungi [13,14,15]. Subsequently, many polyubiquitin genes have been functionally characterized via gene deletion in several fungi including *S. cerevisiae* [6,11,16]; the human pathogen, *Candida albicans* [17]; and the plant pathogenic fungi, *Magnaporthe oryzae *and* Cryphonectria parasitica* [12,18]. For example, deletion of the yeast polyubiquitin gene *UBI4* resulted in reduced resistance to stresses such as high temperature and oxidative stress [6,11,16]. Inactivation of *UBI4 also* affected fungal growth and stress resistance in the human pathogen, *C. albicans* [17]. Furthermore, disruption of the polyubiquitin gene resulted in defective development and pathogenicity in *M. oryzae *and* C. parasitica* [12,18]. However, it is unclear whether the polyubiquitin gene is required for fungal development, stress tolerance, and virulence in entomopathogenic fungi such as *Metarhizium robertsii*.

*Metarhizium robertsii*, as a representative species of entomopathogenic fungi, has been employed as a promising biological control agent against pest insects [19]. However, based on fungal conidia, this biological control agent only accounts for a small percentage of the total insecticide market, which is, at least in part, due to its poor stress resistance in the field [20,21]. In fact, during freight of these fungal products and after application in the field, these conidia may be exposed to environmental stress conditions such as high temperature and ultraviolet (UV) rays [21,22]. Previous studies revealed that high temperature or short-time exposure to simulated solar radiation dramatically reduced the conidial viability in *Metarhizium* [23,24]. Furthermore, previous investigations also demonstrated that genetically engineering *Beauveria bassiana* and *Metarhizium* increased their fungal tolerance to UV radiation and heat, which might be more environmentally feasible in terms of controlling pest insects in the field [25,26,27]. Therefore, more knowledge regarding conidiation and environmental stress tolerance is necessary to genetically engineer a more efficient mycoinsecticide [26].

Previously, we found that the ubiquitin-proteasome pathway was significantly induced by heat stress in *M. robertsii* [28]. Here, the polyubiquitin gene, *MrUBI4* (MAA_02160), an ortholog of the yeast *UBI4* gene, was identified and disrupted via gene deletion in *M. robertsii*. The results show that *MrUBI4* is required for conidiation, conidial germination, and environmental stress tolerance in *M. robertsii.*

## 2. Materials and Methods

### 2.1. Strains and Culture Conditions

The wild-type (WT) strain, *M. robertsii* ARSEF 23, and its mutants were routinely maintained on potato dextrose agar (PDA, 20% potato, 2% dextrose, and 2% agar, *w/v*) plates for 14 days at 25 °C in the dark prior to conidia collection. Subsequently, conidial suspensions were prepared in sterile 0.05% Tween-80 solutions and filtered through non-woven fabric (Yirou, Wuhu, China) to remove mycelia for different experiments. *M. robertsii* transformation mediated by *Agrobacterium tumefaciens* AGL-1 was carried out according to Fang’s protocol [29].

### 2.2. Sequence Resource and Phylogenetic Analysis

Amino acid sequences of the polyubiquitin protein from M. robertsii, S. cerevisiae, Schizosaccharomyces pombe, C. albicans, M. oryzae, Trichoderma parareesei, Fusarium oxysporum, Arabidopsis thaliana, Zea mays, Caenorhabditis elegans, Mus musculus, and Homo sapiens were downloaded from National Center for Biotechnology Information resources (NCBI; http://www.ncbi.nlm.nih.gov/). Phylogenetic analysis was performed using MEGA7 software (http://www.megasoftware.net) [30]. The conserved domain database (CDD, https://www.ncbi.nlm.nih.gov/cdd/) was utilized to search for conserved protein domains. 

### 2.3. Gene Deletion and Complementation

Targeted gene deletion in *MrUBI4* was performed according to our previously described approach based on homologous recombination [31,32]. Primers used for gene deletion and complementation are listed in Table 1. Briefly, the 5’ flanking region (*Eco*RI/*Pst*I (Thermo Scientific, Waltham, MA, USA) with primer set *MrUBI4*-5F/*MrUBI4*-5R) and 3’ flanking region (*Xba*I (Thermo Scientific) with primer set *MrUBI4*-3F/*MrUBI4*-3R) of *MrUBI4* (primer sequences in Table 1) were amplified from genomic DNA (extracted using the Plant Genomic DNA Kit; Tiangen, Beijing, China) using high-fidelity *Taq* DNA polymerase (KOD-Plus-Neo; Toyobo; Osaka, Japan), and subsequently inserted into the binary vector, pDHt-SK-*bar* (kindly provided by Dr. Chengshu Wang; the vector conferred resistance against glufosinate-ammonium), for fungal transformation [29] in order to generate the gene deletion mutant, Δ*MrUBI4* [32]. The transformants, resistant to glufosinate-ammonium (Dr. Ehrenstorfer, Augsburg, Germany; final concentration: 200 μg·mL^−1^), were collected and subsequently screened via genomic PCR using the primer sets, *MrUBI4*-upF/*MrUBI4*-upR (P5/P6), *MrUBI4*-dnF/*MrUBI4*-dnR (P7/P8), *MrUBI4*-F/*MrUBI4*-R (P1/P2), and *bar*-F/*bar*-R (P3/P4) (primer sequences in Table 1). The potential gene mutants were further confirmed via reverse transcription (RT)-PCR using the primer set *MrUBI4*-F/*MrUBI4*-R (P1/P2). The *gpd* (glyceraldehyde 3-phosphate dehydrogenase, MAA_07675) gene was used as the internal control [33].

For mutant complementation, the *MrUBI4* gene was amplified together with its promoter and terminator regions using the primer pair *MrUBI4*CP-5F/*MrUBI4*CP-3R, following which the product was digested with *Spe*I/*Xba*I (Thermo Scientific) and then inserted into the binary vector, pDHt-SK-*ben* (which conferred resistance against benomyl), to transform Δ*MrUBI4* and obtain the complementary strain (Comp). The resulting transformants were picked up via benomyl (Aladdin, Shanghai, China; final concentration: 4 μg·mL^−1^) resistance and verified by PCR and RT-PCR analyses using the primer set, *MrUBI4*-F/*MrUBI4*-R (P1/P2). 

To further identify positive transformants, Southern blot assay was performed as described previously [34,35]. Genomic DNA (10 μg) was digested using *Xho*I (Thermo Scientific) and separated on a 0.7% agarose gel. The corresponding DNA fragments were transferred onto a nylon membrane (Hybond-N^+^ membranes; Amersham Biosciences, Amersham, UK). The 5′ upstream sequence fragment of *MrUBI4* was amplified and labeled with the PCR DIG (digoxigenin) Probe Synthesis Kit (Roche, Basel, Switzerland) for probe synthesis. Hybridization was performed using the DIG-High Prime DNA Labeling and Detection Starter Kit II (Roche) following the manufacturer’s instructions. Signals were visualized via the bioimaging analyzer, Gel Doc XR (Bio-Rad, Hercules, CA, USA). 

### 2.4. Phenotype Assays

For phenotype assays, several experiments were performed with 3 replicates per strain (WT, *∆MrUBI4*, and Comp). The corresponding experiments were also repeated thrice.

For vegetative growth assay, 1 μL aliquots of WT, *∆MrUBI4*, and Comp strain conidial suspensions (1×10^7^ conidia mL^−1^; the same is used in all instances unless specified) were spotted centrally onto PDA, SDAY (Sabouraud dextrose agar with yeast extract: 4% glucose, 1% peptone, and 2% agar plus 1% yeast extract powder, *w/v*), and 1/4 SDAY (amended with 1/4^th^ the amount of nutrients in SDAY) plates. To determine the effects of *MrUBI4* gene deletion on fungal growth, the colony diameter was measured daily after being inoculated for 4 days and maintained at 25 °C in the dark. The colonies were photographed 10 days after incubation.

For sporulation assay, conidial yields were determined as described previously [32]. Briefly, 40 μL aliquots of conidial suspensions were evenly spread on a PDA plate (35 mm diameter) and incubated for 7 and 14 days at 25 °C. The concentration of the conidial suspensions was measured using a hemocytometer and converted to the number of conidia per square centimeter of colony (final concentration: about 10^7^ conidia cm^−2^). 

For conidial germination assay, 10 μL of conidial suspensions were dropped (not spread) onto the center of PDA plates [36]. Conidial germination was observed using a microscope (Olympus BX 51; Tokyo, Japan) at 8, 12, 16, 24, 28, and 36 h after incubation. At least 300 conidia per plate were evaluated, and the germination percentage was calculated based on the number of germinated conidia compared to the 300 counted conidia [36]. A conidium was considered to have germinated when the length of the germ tube reached the same or was longer than the length of the conidium [28].

To determine tolerance to UV, heat, and low temperature stress, the germination rate of conidia was measured [23,24,25]. Briefly, 10 μL aliquots of conidial suspensions were spotted onto the center of PDA plates and exposed to UV-B radiation with a wavelength of 312 nm at 100 μJ·cm^−2^ using a UV crosslinker (UVP HL-2000 HybriLinker; CA, USA) at an energy value set to 1. The plates were incubated at 25 °C for 16 and 24 h after exposure to UV radiation, and conidial germination, for at least 300 conidia per plate, was observed using a microscope (Olympus BX 51) based on our previous report [28]. For the heat and low temperature stress tolerance assay [23,24,25], 1 mL aliquots of conidial suspensions were transferred to 1.5 mL Eppendorf tubes, which were then kept in a water bath at 40 °C for 90 min or in a refrigerator at 4 °C for 90 min. Subsequently, 10 μL of the treated conidial suspensions were inoculated onto PDA plates for 16 and 24 h at 25 °C. Conidial germination, for at least 300 conidia per plate, was then observed using a microscope (Olympus BX 51) based on our previous report [28]. Conidia that were not subject to UV, heat, and low temperature stress tests were also investigated in parallel as described above (conidial germination assay).

For vegetative growth assay under continuous heat stress conditions [37], 1 μL of conidial suspensions were spotted centrally onto PDA plates (90 mm diameter). Similarly, for conidial production assay under continuous heat stress conditions, 40 μL of conidial suspensions were spread evenly onto PDA plates (35 mm diameter). The plates were then incubated at 25 °C (normal growth) or 35 °C (continuous heat stress). The colony diameter was measured daily after being inoculated for 4 days, and conidial yields were investigated after 7- and 14-day incubations as described previously.

For the chemical stress tolerance assay [32], 1 μL of conidial suspensions were pipetted onto the center of PDA plates alone (control) or supplemented with a sensitive concentration (based on [32]) of NaCl (1 M) to induce hyperosmotic stress, H_2_O_2_ (2 mM) and menadione (0.2 mM) to induce oxidative stress, and sodium dodecyl sulfate (SDS; 0.025%) and Congo red (2 mg·mL^−1^) to destabilize cell wall integrity. The mean diameter of each fungal colony was cross-measured under each stress treatment plus control after cultivation at 25 °C for 10 days. Subsequently, the growth inhibition rate was calculated using the following equation: (D_control_ − D_treated_) / D_control_ × 100 (where, D represents the mean colony diameter of the 3 different strains). 

For fungal virulence assay, bioassays were conducted using the last instar larvae of *Galleria mellonella* (RuiQing Bait, Shanghai, China) as described previously [34,35]. Conidia from each strain were applied in normal cuticle infection by dipping the larvae in an aqueous suspension (containing 1 × 10^7^ conidia mL^−1^) for 1.5 min or by injecting a 5-μL conidial suspension (containing 5 × 10^5^ conidia mL^−1^) into the haemocoel of each larva. Each treatment had 3 replicates with 15 insects and all the experiments were repeated thrice. Mortality was examined every 24 h and the median lethal time (LT_50_) was determined and compared via Kaplan-Meier analysis using SPSS (version 16.0, Chicago, IL, USA). 

### 2.5. Transcriptional Profiling Analysis Using Quantitative RT-PCR

Quantitative RT-PCR (qRT-PCR) was performed as described previously [32,38]. To analyze the transcription levels of conidiation-associated genes, 100 μL aliquots of conidial suspensions were evenly spread onto PDA plates and samples was collected after 2.5 days of cultivation (initial period of conidia formation) [39]. To analyze the transcription levels of heat stress-related genes [40], the conidia samples were exposed to 40 °C for 90 min and harvested according to the method mentioned in the heat-stress tolerance assay. To analyze the transcription levels of DNA damage response-related genes [41], the conidia samples were exposed to UV-B radiation and harvested according to the method mentioned in the UV tolerance assay. Conidia not subject to heat stress were also simultaneously investigated as controls. Total RNA was isolated using TRIzol reagent (Invitrogen, Foster City, CA, USA), and complementary DNA (cDNA) was synthesized from 100 ng of total RNA using the PrimeScript™ RT reagent Kit with gDNA Eraser (TaKaRa, Dalian, China). cDNA samples (10× dilution) were then used as templates for quantitative real-time PCR (qPCR), and all the reactions were run in triplicate. qPCR was performed via the CFX96™ Real-Time PCR System (Bio-Rad) using specific primer sets (Appendix A) and SYBR^®^ Premix Ex Taq™ II (TaKaRa Dalian) according to the manufacturer’s instructions. The threshold cycle (C_T_) was determined using the default threshold settings. *gpd* was used as an internal control. Relative transcription levels of the genes related to conidiation and heat stress in *∆MrUBI4* and Comp strains were calculated using the 2^-ΔΔCt^ method [42] by comparing them to the standard transcription levels of WT. 

### 2.6. Statistical Analysis

All data are expressed as mean ± standard deviation (SD) of 3 independent experiments. Statistical analyses were performed via GraphPad Prism version 6.0 (GraphPad, Inc., La Jolla, CA, USA) using one-way analysis of variance (ANOVA). Furthermore, Tukey’s test was used to compare differences between the mean values. *p* < 0.05 was considered to be statistically significant.

## 3. Results

### 3.1. Identification and Characteristics of MrUBI4 from *Metarhizium robertsii*

To identify orthologs of yeast polyubiquitin in *M. robertsii*, the protein sequences of *S. cerevisiae* UBI4 (YLL039C) were used as queries for BLASTP analyses in NCBI employing the Basic Local Alignment Search Tool (http://blast.ncbi.nlm.nih.gov/blast.cgi) against the *M. robertsii* genome database [43]. The results identified a single copy of the polyubiquitin gene (MAA_02160) in *M. robertsii* that encoded a 305-aa protein showing 79.8% amino acid identity to *S. cerevisiae* UBI4. Therefore, it was designated as *MrUBI4*. 

A phylogenetic analysis of MrUBI4 and the polyubiquitin orthologs in the other organisms showed that MrUBI4 was highly conserved among fungi, plants, and insects (Figure 1A). Further analysis of the amino acid sequence alignment of polyubiquitin showed that variability in the repeat number of monomeric ubiquitin among different species resulted in differences in sequence identity (Figure 1B). Of these, *MrUBI4* encoded a protein with 4 monomeric ubiquitin repeats.

### 3.2. Targeted Disruption of MrUBI4

To further assess the physiological roles of *MrUBI4*, the targeted gene disruption vector, pDHt-SK-*bar*-*MrUBI4*, was introduced into the WT strain in order to obtain a gene deletion strain by homologous replacement (Appendix A). In the present study, the *MrUBI4* coding region plus the partial sequence upstream of the coding region (1118 bp) was replaced with the glufosinate-ammonium-resistance gene (*bar* cassette, 942 bp). Moreover, Comp strains were also generated based on ∆*MrUBI4* by introducing the gene-coding region plus promoter of *MrUBI4*. As expected, positive transformants were confirmed via genomic PCR, RT-PCR, and Southern blot analysis (Appendix A). Initially, only 2 recombinant fragments with 1100 and 1010 bp were amplified by primer sets P5/P6 and P7/P8 in ∆*MrUBI4*, and not in WT (Appendix A). Further genomic PCR and RT-PCR analysis showed that a 462-bp fragment, corresponding to the partial *MrUBI4* gene, was detected in WT and Comp strains, but not in ∆*MrUBI4* (Appendix A). Genomic PCR analysis also showed that the *bar* gene fragment (434 bp) was identified in ∆*MrUBI4* and Comp strains, whereas no fragment was found in WT (Appendix A). Furthermore, the benomyl-resistance gene (*ben*) fragment (301 bp) was only partially identified in the Comp strain, whereas no band was detected in WT and ∆*MrUBI4* (Appendix A). In addition, the results of the Southern blot analysis indicated the integration of a single copy of the disruption cassette at the *MrUBI4* locus in the ∆*MrUBI4* strain (Appendix A).

### 3.3. Effects of MrUBI4 Deletion on Hyphal Growth

To determine whether *MrUBI4* was associated with hyphal growth in *M. robertsii*, we evaluated the growth of *∆MrUBI4* on PDA, SDAY, and 1/4 SDAY media. The *∆MrUBI4* strain showed slightly smaller colony diameters than the WT and Comp strains on all different media types under the same culture conditions (25 °C in the dark for 10 days (Figure 2A)). This was also supported by statistical analysis (Figure 2B) and continuous observation during the 4–14-day cultivation period (Figure 2C). Overall, these results suggest that MrUBI4 is dispensable for hyphal growth in *M. robertsii*.

### 3.4. MrUBI4 is Important for Conidiation and Conidial Germination

To test whether *MrUBI4* played a role in conidiation, the sporulation ability of different strains on PDA plates was measured 7- and 14-days post-incubation. The results showed that at 14 days of growth, conidia production in the WT, *∆MrUBI4*, and Comp strains were 5.96 × 10^7^ ± 0.31 × 10^7^, 2.33 × 10^7^ ± 0.21 × 10^7^, and 5.64 × 10^7^ ± 0.30 × 10^7^ conidia cm^−2^, respectively (Figure 3A). These results indicated that the *∆MrUBI4* conidia yield was significantly decreased by 45% (*p* < 0.01) and 61% (*p* < 0.01) at 7- and 14-days post-incubation respectively, when compared to the WT strain (Figure 3A). Furthermore, the expression of several conidiation-related genes at the transcription level was also examined via qRT-PCR analysis. The results showed that the *brlA*, *abaA*, *wetA*, and *flbD* transcript levels were significantly lower in the *∆MrUBI4* strain than in the WT strain (*p* < 0.01; Figure 3C), which indicated that *MrUBI4* was involved in the regulation of conidiation-related gene expression. Furthermore, *flbA*, *flbB*, *flbC*, *fluG*, *vosA*, *StuA*, *cag8*, *mero*-*Fus3*, *Pks1*, and *Mlac1* expression levels did not display significant differences between the *∆MrUBI4* and control strains.

Conidial germination on PDA medium was also observed to be significantly delayed in *∆MrUBI4* when compared to WT and Comp (Figure 3B,D). While only about 16% of the conidia from *∆MrUBI4* germinated at the 12-h mark, 40% of the conidia from WT and Comp had germinated in the same timeframe. After incubating for 24 h, the percentage of conidial germination in the WT and Comp strains was about 88%, whereas the same in *∆MrUBI4* was only 58% approximately (Figure 3B). Finally, the difference in conidial germination between the WT, *∆MrUBI4*, and Comp strains gradually diminished after 36 h of incubation. Thus, our findings demonstrate that *MrUBI4* is important in conidial production and germination.

### 3.5. Contribution of MrUBI4 to Environmental Stress Tolerance

To investigate whether *∆MrUBI4* exhibited any defects under different environmental stress conditions, the conidia of WT and mutants were first exposed to a sensitive dose of UV, following which the conidia germination rates were examined after incubation for 16 and 24 h. The results showed that the germination rates in WT, *∆MrUBI4*, and Comp decreased under UV stress when compared to normal conditions. However, the germination rates of the ∆*MrUBI4* strain decreased more significantly under UV stress when compared to that in the WT and Comp strains (Figure 4A,B). For example, compared to the WT, *∆MrUBI4* germination rates at 24 h decreased by 47.2% (*p* < 0.01) under UV stress, but only 31.0% (*p* < 0.01) under normal conditions (Figure 4A,B). Similar results were also found when the conidia of different strains were heat-stressed at 40 °C for 90 min in a water bath. For example, compared to the WT, *∆MrUBI4* germination rates at 24 h decreased by 46.8% (*p* < 0.01) under heat stress, but only 31.0% (*p* < 0.01) under normal conditions (Figure 4A,B). However, the WT, *∆MrUBI4*, and Comp strain germination rates under low temperature (4 °C) stress were similar to those under normal conditions (Figure 4A,B), which is consistent with the fact that this biocontrol agent based on conidia is usually transported using the cold chain method. Interestingly, after 40 °C heat treatment, we found that the mRNA expression level of several genes, such as *hsp*, *tps*, *mpd*, and *tpp*, decreased by more than 50% in the *∆MrUBI4* strain when compared to the WT and Comp strains (*p* < 0.01; Figure 4C). Whereas after UV treatment, the mRNA expression level of only one gene (*uve-1*), decreased by approximately 50% in the ∆*MrUBI4* strain, compared to the control strains (*p* < 0.01; Figure 4C). Under normal conditions, the relative expression of most of the stress response genes was either unchanged (such as *tps*, *mpd* and *tpp*) or upregulated (such as *hsp30*) in *∆MrUBI4* when compared to those in WT and Comp (Figure 4D). Furthermore, the vegetative growth and conidial production assays under continuous heat stress conditions showed that the vegetative growth of the ∆*MrUBI4* strain was dramatically inhibited (Figure 5A) and that its conidial yield was significantly reduced (Figure 5B). These findings suggest that *MrUBI4* contributes to the UV and heat stress response of the fungus.

Additionally, to investigate whether *∆MrUBI4* exhibited any defects under different chemical stress conditions, colony diameters of the WT and mutant strains were measured on plates containing NaCl, H_2_O_2_, menadione, SDS, and Congo red (Appendix A), and the data was presented as growth inhibition rates (Appendix A). Our results found that the growth inhibition rate of *∆MrUBI4* in the presence of stress-inducing chemicals was not significantly different from that of the control strains. These results indicate that *MrUBI4* is not involved in the regulation of chemical stress in *M. robertsii*.

### 3.6. MrUBI4 Has No Effect on Fungal Virulence

To determine whether *MrUBI4* was involved in the pathogenicity of *M. robertsii*, conidial suspensions of the WT, *∆MrUBI4*, and Comp strains were used to infect *G. mellonella* via topical immersion or injection. Mortality was observed daily over a 14-day period, and our results showed no significant difference in LT_50_ (mean lethal time for 50% mortality) values between the *∆MrUBI4* and control strains, regardless of whether topical inoculation or injection was used (Appendix A). These results demonstrate that *MrUBI4* is not involved in *M. robertsii* pathogenicity.

## 4. Discussion

Many previous studies have established that the polyubiquitin gene likely provides the main supply of cellular monomeric ubiquitin protein in response to developmental and environmental stimuli in eukaryotes [5,6,7,8]. In yeast, the polyubiquitin gene has been reported to encode multiple monomeric ubiquitin repeats in tandem, which is transcribed and translated into a multi-unit ubiquitin precursor that is subsequently cleaved by specific DUBs to yield mature ubiquitin monomers [6,8]. Polyubiquitin gene expression could also produce a rapid response under stress conditions when compared to monomeric ubiquitin fusion gene expression [8]. Similar to the *S. cerevisiae UBI4* that has 5 repeats [6,8], we found that *MrUBI4* contained 4 monomeric ubiquitin repeats, which was comprised of 76 amino acids. The polyubiquitin gene of *M. oryzae* was also found to contain 4 monomeric ubiquitin repeats [8,12]. Although the homologous proteins of MrUBI4 have a similar number of monomeric ubiquitin repeats, functional differences may exist between them. Due to the different regulatory elements in the polyubiquitin genes from these fungi, expression of the same coding region ultimately resulted in different functions that responded differently to different environmental conditions [44]. For example, the yeast polyubiquitin gene, *UBI4*, was reported to be involved in stress response in *S. cerevisiae* [6,11,16]. In *M. oryzae*, disruption of the polyubiquitin gene resulted in defective growth, development, and pathogenicity [12]. Relatively little is known about the biological functions of the polyubiquitin gene in insect pathogens. Therefore, in this study, the role of *MrUBI4* is explored in the filamentous entomopathogenic fungus, *M. robertsii*.

The polyubiquitin gene in fungi has been reported to be involved in stress responses such as heat stress [6,14,16,17]. Consistent with previous studies in yeast, *C. albicans*, and *C. parasitica* [6,17,18], the *MrUBI4* mutant showed decreased resistance to high temperatures when compared to control strains. Previously, we found that the ubiquitin-proteasome pathway was significantly induced by heat stress in *M. robertsii* [28]. In the present study, further analysis of the polyubiquitin gene-deletion strains after heat and UV stress induction suggests that the ubiquitin-related gene (*MrUBI4*) plays an important role in the environmental stress response in *M. robertsii* (Figure 4 and Figure 5). Intriguingly, a significant increase of sensitivity to heat shock was accompanied with reduced transcript levels of genes related to heat-shock protein (*hsp*), trehalose synthesis (*tps*, *tpp*, and *nth*), and mannitol accumulation (*mpd*). Accumulation of heat shock proteins, trehalose, and mannitol have also been reported to contribute to conidial stress tolerance in *M. robertsii* and other filamentous fungi [25,45,46,47]. Unfortunately, only a few reports detail the role of the polyubiquitin gene from other fungi with respect to chemical stress response. In this study, we did not find any obvious difference in chemical stress tolerance between the mutant and control strains. Based on these results, we speculated that the polyubiquitin gene was mainly involved in the regulation of responses to environmental stress.

Furthermore, the conidiation assays showed that the conidia yield in *∆MrUBI4* was significantly reduced when compared to the control. Significantly reduced conidia yields were also observed in the polyubiquitin gene-deletion strains of *M. oryzae* [12] and *cpubi4* (homolog gene of yeast *UBI4*)-deletion strains of *C. parasitica* [18]. Accordingly, our qPCR results showed that the transcript levels of several conidiation-related genes, including *brlA*, *abaA*, *wetA*, and *flbD*, were significantly decreased in *∆MrUBI4*, thus indicating that *MrUBI4* might be involved in the regulation of asexual sporulation. Previously, *brlA*, *abaA*, and *wetA* were reported to be involved in central regulatory pathways associated with conidiation; thus, playing important roles in the regulation of asexual sporulation in different filamentous fungi [48,49,50]. Conidial germination was also observed to be significantly delayed in the *MrUBI4* deletion strains, which was consistent with the recent study in *M. oryzae* that showed that the disruption of *MoRad6* (a representative ubiquitin conjugating enzyme E2 involved in the ubiquitination-related pathway) led to severe delay in conidial germination [51]. Altogether, these results suggest that *MrUBI4* is required for conidiation and conidial germination.

In our study, the pathogenicity assays showed that *MrUBI4* disruption did not lead to reduction in fungal virulence. However, the polyubiquitin gene is required for pathogenicity in the plant pathogens, *M. oryzae* and *C. parasitica* [12,18]. Therefore, we speculated that the polyubiquitin gene may be involved in different fungal virulence pathways in plants and insects, as fungal virulence has been reported to be regulated by different mechanisms in plant and insect pathogens [19]. Moreover, many entomopathogenic fungi, including *Metarhizium* and *Beauveria*, are also plant endophytes [19,52,53]. As an endophyte, *M. robertsii* has the ability to colonize plants endophytically and stimulate plant root development [54,55]. Considering the endophytic role of *M. robertsii*, we speculated that the polyubiquitin gene, *MrUBI4*, could play a role in plant colonization.

Collectively, our data suggests that the polyubiquitin gene, *MrUBI4*, is involved in the regulation of conidiation, conidial germination, and heat and UV stress in *M. robertsii*. Further studies need to be performed to analyze the ubiquitination level of specific target proteins related to heat and UV stress tolerance in polyubiquitin gene-deletion strains, which may be helpful in genetically enhancing *M. robertsii*.

## Figures and Tables

**Figure 1 genes-10-00412-f001:**
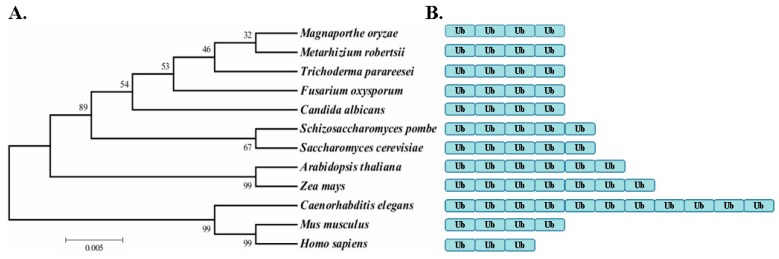
Phylogenetic analysis of *MrUBI4* and the polyubiquitin orthologs from various fungi, plants, and animals. **(A)** The phylogenetic tree was generated via the MEGA7 software using default settings (neighbor-joining method). The amino acid sequences of polyubiquitin that were used in this analysis are as follows: *Magnaporthe oryzae* (EHA54394.1), *Metarhizium robertsii* (MAA_02160), *Trichoderma parareesei* (OTA03154.1), *Fusarium oxysporum* (ENH62031.1), *Candida albicans* (CAA90901.1), *Arabidopsis thaliana* (CAB81074.1), *Saccharomyces cerevisiae* (YLL039C), *Schizosaccharomyces pombe* (SPBC337.08c), *Zea mays* (NP_001316595.1), *Caenorhabditis elegans* (F25B5.4/WBGene00006727), *Mus musculus* (MGI:98888), and *Homo sapiens* (ENSG00000170315). **(B)** The variant number of monomeric ubiquitin repeats in the polyubiquitin protein of different species. Box with Ub in blue represents 1 monomeric ubiquitin repeat.

**Figure 2 genes-10-00412-f002:**
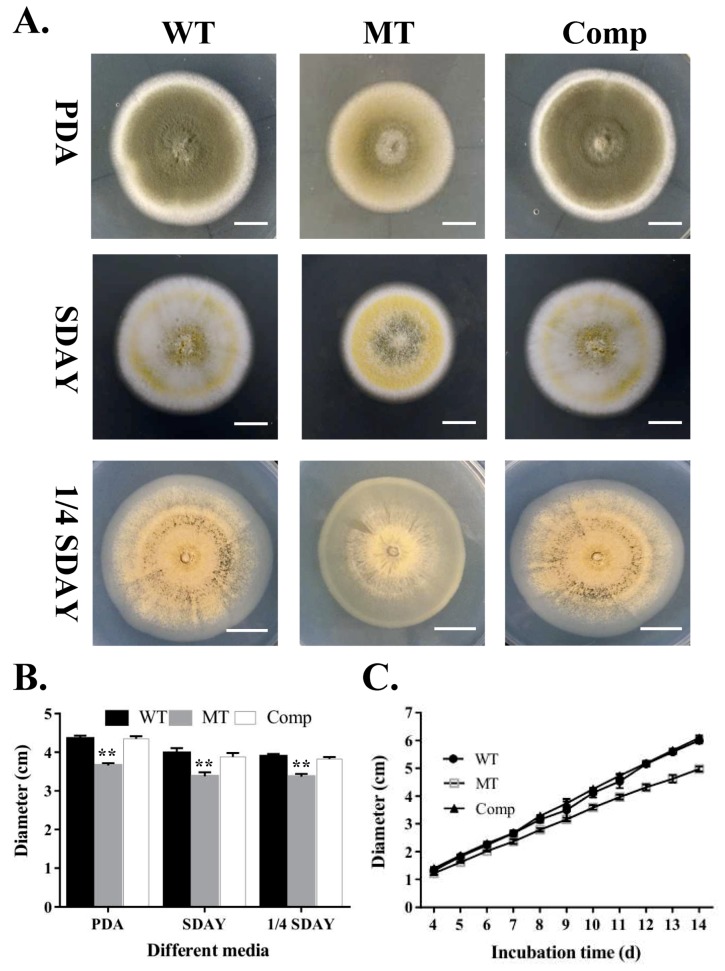
Effects of *MrUBI4* deletion on vegetative growth. **(A)** Colony phenotyping of the WT and mutant strains on PDA, SDAY, and 1/4 SDAY plates after growing for 10 days at 25 °C. Scale: 1 cm. **(B)** Fungal colony diameters of 3 different strains on PDA, SDAY, and 1/4 SDAY plates after growing for 10 days at 25 °C. **(C)** Fungal growth of 3 different strains during the 4-14 day cultivation period on PDA plates at 25 °C, and the growth rates of the ∆*MrUBI4* strain reduced significantly (Tukey’s test, *p* < 0.01) after 4 days post-inoculation. **, *p* < 0.01. WT: wild-type strain, MT: ∆*MrUBI4* strain, Comp: complementary strain.

**Figure 3 genes-10-00412-f003:**
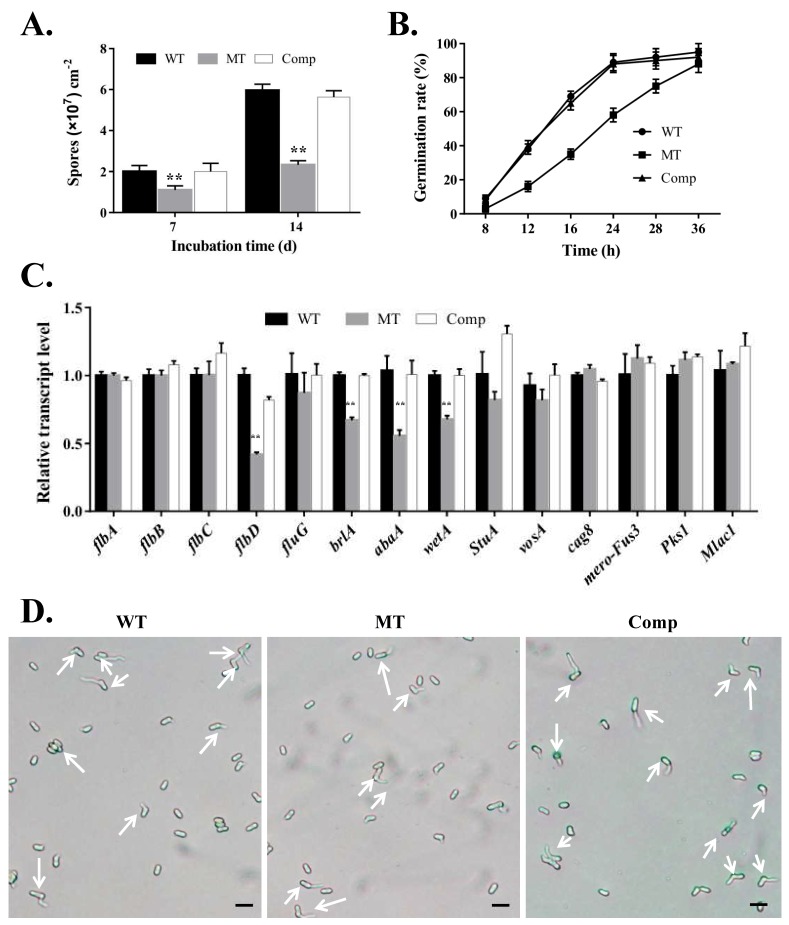
*MrUBI4* is required for conidiation and conidial germination. **(A)** Conidial yields of different strains cultured on PDA plates for 7 and 14 days. **(B)** Kinetics of conidial germination. Conidial germination rate 8-36 h after inoculation is presented. **(C)** Relative expression analysis of conidiation-related genes in the 2.5-day-old PDA cultures. **(D)** Representative images of conidial germination 12 h after inoculation. Arrowhead (white) depicts germinated conidia. Scale: 10 μm. **, *p* < 0.01. WT: wild-type strain, MT: ∆*MrUBI4* strain, Comp: complementary strain.

**Figure 4 genes-10-00412-f004:**
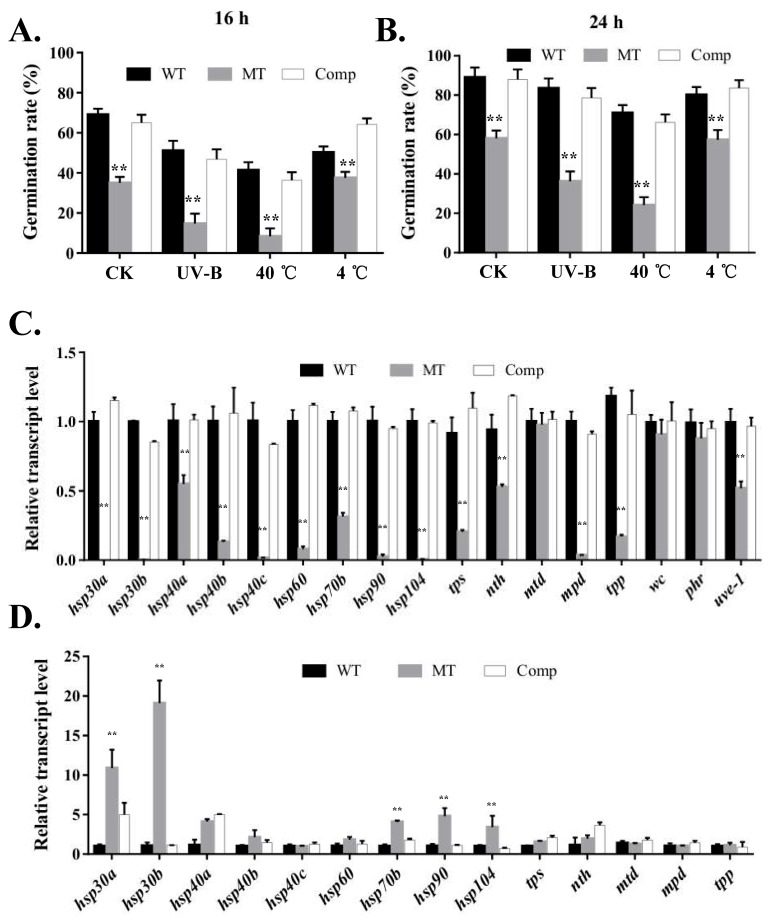
Spore survival assays of different *M. robertsii* strains following UV-B, heat and low temperature treatment. Germination rate of conidia from the 3 strains at 16 h **(A)** and 24 h **(B)** after UV-B, heat and low temperature exposure. Conidia of the 3 strains (WT, ∆*MrUBI4*, and Comp) were either UV-B irradiated at 100 μJ·cm^−2^ or heated at 40 °C (and low temperature at 4 °C) for 90 min. Germination rates were determined after inoculation on PDA plates at 25 °C for 16 and 24 h. **(C)** Relative expression analysis of heat stress- and DNA damage response-related genes in conidia under thermal- and UV-treated stress. **(D)** Relative expression analysis of heat stress-related genes in conidia under normal conditions (25 °C for cultivation). **, *p* < 0.01. WT: wild-type strain, MT: ∆*MrUBI4* strain, Comp: complementary strain.

**Figure 5 genes-10-00412-f005:**
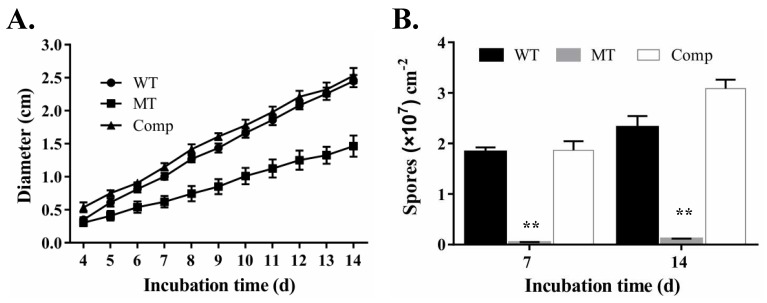
Assays for vegetative growth and conidial production under continuous heat-stress conditions. **(A)** Vegetative growth of different strains during the 4–14-day cultivation period on PDA plates under heat stress (35 °C), and the growth rates of the ∆*MrUBI4* strain decreased significantly (Tukey’s test, *p* < 0.01) after 4 days post-inoculation. **(B)** Conidial production of different strains cultured on PDA plates for 7 and 14 days under heat stress (35 °C). **, *p* < 0.01. WT: wild-type strain, MT: ∆*MrUBI4* strain, Comp: complementary strain.

**Table 1 genes-10-00412-t001:** Primers used for gene deletion and complementation.

Gene	Primer Name	Sequence (5’- 3’)	Notes
*MrUBI4*	*MrUBI4*-5F	GGAATTCGAGCAAGACAAGCCAACG, *Eco*RI	For construction of gene disruption vector
	*MrUBI4*-5R	AACTGCAGAAGAAAGCAGGGTCAAGAT, *Pst*I	
	*MrUBI4*-3F	GCTCTAGACAGTAGTTGATTGGACGATG, *Xba*I	
	*MrUBI4*-3R	GCTCTAGACTGGGAGTAAAGTGGAAGAT, *Xba*I	
	*MrUBI4*-upF(P5)	GTGGCTGTCATCAGGAGTTT	PCR identification of *MrUBI4* deletion transformants
	*MrUBI4*-upR(P6)	GGCATTCATTGTTGACCTCC	
	*MrUBI4*-dnF(P7)	GTTTCTGGCAGCTGGACTTC	
	*MrUBI4*-dnR(P8)	AGCGTGGACAGACTTTGATTT	
	*MrUBI4*CP-5F	GGACTAGTGGGTGGACTGGAGGTA, *Spe*I	For gene complementation
	*MrUBI4*CP-3R	GCTCTAGATAGGAATCGAACGCAGTT, *Xba*I	
	*MrUBI4*-F(P1)	GGAAGTCACTAACAATCCCACG	Genomic PCR and RT-PCR analysis
	*MrUBI4*-R(P2)	AAGTCGCAGGACAAGGTG	
*bar*	*bar*-F(P3)	TCGTCAACCACTACATCGAGAC	Genomic PCR analysis
	*bar*-R(P4)	GAAGTCCAGCTGCCAGAAAC	
*ben*	*ben*-F	GGTAACTCCACCGCCATCCA	Genomic PCR analysis
	*ben*-R	GCAGGGTATTGCCTTTGGACTT	
*gpd*	*gpd*-F	GACTGCCCGCATTGAGAAG	RT-PCR analysis
	*gpd*-R	AGATGGAGGAGTTGGTGTTG	
	*UBI4-probe-F*	TGATAAGGACGGCGGTTTG	For probe synthesis
	*UBI4-probe-R*	CCGAAGTAGGCAGCACGAT	

**Notes:** Gray underlined boxes indicate the restriction enzyme sites.

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
