# Peer review of "The Polyubiquitin Gene MrUBI4 Is Required for Conidiation, Conidial Germination, and Stress Tolerance in the Filamentous Fungus Metarhizium robertsii"

_genes, 2019, doi:10.3390/genes10060412_

Round 1

Reviewer 1 Report

General comments:

In this manuscript, the authors characterized the role of polyubiquitin gene MrUBI4 in M. robertsii via genetic deletion studies. They demonstrated that MrUBI4 deletion resulted in developmental defects and showed increased sensitivity under various stress conditions. Although this work is potentially interesting, the authors need to provide more convincing evidence and thorough interpretation of their data to improve the manuscript. The descriptions of results are sometimes too concise to understand the author’s work. Furthermore, some of the experimental results do not fully support the author’s conclusions. The following points need to be addressed.

Major comments:

In Figure 1, the authors should note that there are two polyubiquitin genes in M. musculus and H. sapiens. Are there any reasons that they showed only one of the two polyubiquitin genes in mammals?

Is it possible for the authors to show the mRNA and protein levels of ubiquitin when MrUBI4 is deleted (MT strain) and compare them to those levels in WT and Comp strain? How about those levels between WT and Comp strain? Was it fully compensated in Comp strain? Unfortunately, this reviewer could not access to the supplementary data.

In Figure 4C, as the authors showed the relative mRNA levels compared to WT strain for each gene, the value of mRNA levels for each gene in WT strain should be 1. However, some of them are not, and these need to be explained.

Data in Figure 4C were generated under heat stress and those in Figure 4D were generated under normal conditions. To this reviewer, it seems that several hsp gene expression levels already increased under normal conditions, and it was not further increased (or increased to a lesser extent than WT strain) under heat stress conditions in MT strain. While, in WT strain, their expression levels were very low under normal conditions, and significantly increased under heat stress conditions. Therefore, it may not be appropriate to state that gene expression levels “decreased” in MT strain under heat stress conditions. To address this problem, the authors need to compare the raw data in Figure 4C and 4D; and may want to plot the data in the same graph.

Minor comments:

For qRT-PCR data, the authors need to indicate whether their data are normalized to any internal controls.

In Figure 2A, the authors need to include the scale bars.

In Figure 2C, it is hard to distinguish among WT, MT, and Comp strain. The authors may want to change MT strain to open squares instead of solid squares.

In figure 3D, is it possible to provide higher resolution images, too? It is hard to notice the difference in these low resolution of images.

Author Response

Authors' Response to Reviewers' Comments

*********************************************************************

Reviewer 1

General comments:

In this manuscript, the authors characterized the role of polyubiquitin gene MrUBI4 in M. robertsii via genetic deletion studies. They demonstrated that MrUBI4 deletion resulted in developmental defects and showed increased sensitivity under various stress conditions. Although this work is potentially interesting, the authors need to provide more convincing evidence and thorough interpretation of their data to improve the manuscript. The descriptions of results are sometimes too concise to understand the author’s work. Furthermore, some of the experimental results do not fully support the author’s conclusions. The following points need to be addressed.

Response:

Thank you for your observations.

For your suggestion: “Furthermore, some of the experimental results do not fully support the author’s conclusions”, we have revised the RESULTS section “3.3. Deletion of MrUBI4 affects hyphal growth”, and the sentence was added “Overall, these results suggest that MrUBI4 is dispensable for hyphal growth in M. robertsii.

Major comments:

In Figure 1, the authors should note that there are two polyubiquitin genes in M. musculus and H. sapiens. Are there any reasons that they showed only one of the two polyubiquitin genes in mammals?

Response:

Yes. We showed only one (Ubb) of the two polyubiquitin genes in mammals, just because the repeat number of Ubb is closer to the filamentous fungus, including Metarhizium robertsii.

Is it possible for the authors to show the mRNA and protein levels of ubiquitin when MrUBI4 is deleted (MT strain) and compare them to those levels in WT and Comp strain? How about those levels between WT and Comp strain? Was it fully compensated in Comp strain? Unfortunately, this reviewer could not access to the supplementary data.

Response:    

Yes.

In fact, the mRNA level of polyubiquitin (MrUBI4) was examined by semi-quantitative RT-PCR analysis (see details in Figure S1C). Our results show that the transcripts of MrUBI4 are completely undetectable in the ∆MrUBI4 gene-deletion mutant; whereas it is detectable in the WT and Comp strains, when gpd was used as an internal control (Figure S1C).

The revised supplementary data were also provided by E-mail for the editors and reviewers.

In Figure 4C, as the authors showed the relative mRNA levels compared to WT strain for each gene, the value of mRNA levels for each gene in WT strain should be 1. However, some of them are not, and these need to be explained.

Response:   

Thank you for your suggestion.

According to the classical reference for real-time quantitative PCR (qPCR) analysis, if the value of mRNA levels for each gene in untreated sample (or WT strain) is set to 1, and the results for untreated sample (or WT strain) would be presented as 1 without any error (using the standard deviation), which is similar with the recent report in Metarhizium acridum (Figure 3c, f and Figure 4d in REF 1; Figure 5 in REF 2).

Alternatively, to display the corresponding data with error (using the standard deviation), CT values for different non-replicated samples were carried through the entire 2 -∆∆CT calculation before averaging (Figure 2 in [41]). Using this analysis, the value of the mean fold change for untreated sample (or WT strain) should be very close to 1 (some of them may not be 1), which is shown in this study (also including our recent work [REF 3]).

References

41.       Livak, K.J.; Schmittgen, T.D. Analysis of relative gene expression data using real-time quantitative PCR and the 2 -∆∆CT method. Methods 2001, 25, 402-408, doi:10.1006/meth.2001.1262.

REF 1.  Du, Y.; Jin, K.; Xia, Y. Involvement of MaSom1, a downstream transcriptional factor of cAMP/PKA pathway, in conidial yield, stress tolerances, and virulence in Metarhizium acridum. Appl. Microbiol. Biotechnol. 2018, 102, 5611-5623, doi:10.1007/s00253-018-9020-7.

REF 2.  Xie, M.; Xia, Y.; Cao, Y. The Rab GTPase activating protein Gyp2 contributes to UV stress tolerance in Metarhizium acridum. World J. Microbiol. Biotechnol. 2018, 34, 78, doi:10.1007/s11274-018-2457-0.

REF 3.   Wang, Z.; Jiang, Y.; Li, Y.; Feng, J.; Huang, B. MrArk1, an actin-regulating kinase gene, is required for endocytosis and involved in sustaining conidiation capacity and virulence in Metarhizium robertsiiAppl. Microbiol. Biotechnol.  2019, DOI: 10.1007/s00253-019-09836-6.

Data in Figure 4C were generated under heat stress and those in Figure 4D were generated under normal conditions. To this reviewer, it seems that several hsp gene expression levels already increased under normal conditions, and it was not further increased (or increased to a lesser extent than WT strain) under heat stress conditions in MT strain. While, in WT strain, their expression levels were very low under normal conditions, and significantly increased under heat stress conditions. Therefore, it may not be appropriate to state that gene expression levels “decreased” in MT strain under heat stress conditions. To address this problem, the authors need to compare the raw data in Figure 4C and 4D; and may want to plot the data in the same graph.

Response:  

Thank you for your suggestion.

Using real-time quantitative PCR and the 2 -∆∆CT method, the expression levels for gene of interest were evaluated, only for the gene expression level in relative quantification [41].

According to the recent report in fungal insect pathogen including Metarhizium, the expression levels of corresponding genes among different strains were usually investigated under treated condition [40-41, REF 1-3] (for example, Data in Figure 4C were obtained under heat stress), because the 2 -∆∆CT method is just for presenting gene expression level in relative quantification, not absolute quantification. Therefore, it is not required to provide and analyze the Data in Figure 4D under normal (or untreated) conditions. Here, Data in Figure 4D were generated under normal conditions, which was conducted at the request of another reviewer.  

Furthermore, differences of phenotype and gene expression in MT strains were also obtained by comparison with the control strains (for example, effects of MrUBI4 deletion on phenotypic change and gene expression are all relative to those in the WT and Comp strains).

References

40.       Wang, J.J.; Cai, Q.; Qiu, L.; Ying, S.H.; Feng, M.G. The histone acetyltransferase Mst2 sustains the biological control potential of a fungal insect pathogen through transcriptional regulation. Appl. Microbiol. Biotechnol. 2018, 102, 1343-1355, doi:10.1007/s00253-017-8703-9.

41.       Livak, K.J.; Schmittgen, T.D. Analysis of relative gene expression data using real-time quantitative PCR and the 2 -∆∆CT method. Methods 2001, 25, 402-408, doi:10.1006/meth.2001.1262.

REF 1.  Du, Y.; Jin, K.; Xia, Y. Involvement of MaSom1, a downstream transcriptional factor of cAMP/PKA pathway, in conidial yield, stress tolerances, and virulence in Metarhizium acridum. Appl. Microbiol. Biotechnol. 2018, 102, 5611-5623, doi:10.1007/s00253-018-9020-7.

REF 2.  Xie, M.; Xia, Y.; Cao, Y. The Rab GTPase activating protein Gyp2 contributes to UV stress tolerance in Metarhizium acridum. World J. Microbiol. Biotechnol. 2018, 34, 78, doi:10.1007/s11274-018-2457-0.

REF 3.   Wang, Z.; Jiang, Y.; Li, Y.; Feng, J.; Huang, B. MrArk1, an actin-regulating kinase gene, is required for endocytosis and involved in sustaining conidiation capacity and virulence in Metarhizium robertsiiAppl. Microbiol. Biotechnol.  2019, DOI: 10.1007/s00253-019-09836-6.

Minor comments:

For qRT-PCR data, the authors need to indicate whether their data are normalized to any internal controls.

Response:   

Yes.

The gpd (glyceraldehyde 3-phosphate dehydrogenase, MAA_07675) gene was used as the internal control for semi-quantitative RT-PCR and qRT-PCR analysis. The species-specific primers were the same for in semi-quantitative RT-PCR and qRT-PCR analysis according the previous report [33] (Table 1).

References

33.       Fang, W.G.; Bidochka, M.J. Expression of genes involved in germination, conidiogenesis and pathogenesis in Metarhizium anisopliae using quantitative real-time RT-PCR. Mycol. Res. 2006, 110, 1165-1171, doi:10.1016/j.mycres.2006.04.014.

In Figure 2A, the authors need to include the scale bars. 

Response:  

We apologize for this oversight. The corresponding scale bars were added in the revised Figure 2A.

In Figure 2C, it is hard to distinguish among WT, MT, and Comp strain. The authors may want to change MT strain to open squares instead of solid squares.

Response:  

Thank you for your suggestion, and we have changed MT strain to open squares.

In figure 3D, is it possible to provide higher resolution images, too? It is hard to notice the difference in these low resolution of images.

Response:  

Thank you for your suggestion. We have revised the image contrast for distinguishing the difference of germinated and ungerminated conidia.

Reviewer 2 Report

Wang et al. examined the effects of deleting MrUBI4 gene in Metarhizium robertsii and found that delayed conidial germination, significantly decreased conidial yields,  increased sensitivity to ultraviolet (UV) and heat-shock 25 stress, while has no effect on fungal virulence. These findings may help design more effective anti-insect agents with modified fungi strains to tolerate higher environmental stresses. Overall the manuscript is well written, experiments were designed and analyzed to support the major conclusions. However, there are some concerns need to be addressed before publication.

Major concerns: The whole manuscript is based on Metarhizium robertsii strains with MrUBI4 deletion and complementation. Thus,

(1)   Given that even in yeast there are four copies of ubiquitin genes, it will be necessary to indicate clearly how many ubiquitin expressing genes in Metarhizium robertsii.

(2)   Although genomic DNA has been examined for gene deletion, western blotting would be necessary to confirm the loss of ubiquitin expression, given to multiple copies of ubiquitin are present.

(3)   Ubiquitin expression levels in Comp strains to WT strain would need to be examined.

Minor concerns:

1.     Fig 2A: scale bars are needed.

2.     Fig 2C: statistical analyses are needed.

3.     Fig 4C: to understand why ubiquitin deletion affects UV sensitivity, DNA damage response genes would need to be examined.

4.     Fig 5A: statistical significance is needed.

5.     The reviewer suggests the authors to seek help from English Professionals to eliminate typos and grammar mistakes in the manuscript.

Author Response

Authors' Response to Reviewers' Comments

*********************************************************************

Reviewer 2

Comments and Suggestions for Authors

Wang et al. examined the effects of deleting MrUBI4 gene in Metarhizium robertsii and found that delayed conidial germination, significantly decreased conidial yields,  increased sensitivity to ultraviolet (UV) and heat-shock 25 stress, while has no effect on fungal virulence. These findings may help design more effective anti-insect agents with modified fungi strains to tolerate higher environmental stresses. Overall the manuscript is well written, experiments were designed and analyzed to support the major conclusions. However, there are some concerns need to be addressed before publication.

Major concerns:

The whole manuscript is based on Metarhizium robertsii strains with MrUBI4 deletion and complementation. Thus,

(1)   Given that even in yeast there are four copies of ubiquitin genes, it will be necessary to indicate clearly how many ubiquitin expressing genes in Metarhizium robertsii.

Response:  

Thank you for your suggestion. There may be misunderstandings for the polyubiquitin gene.

According to the reference [6-8], the polyubiquitin gene is a unique, highly conserved open reading frame composed solely of tandem repeats (that is, the polyubiquitin gene is a unique gene with several repeat in tandem, not multiple copies). In yeast, there are one polyubiquitin gene, UBI4, encoding multiple ubiquitin units in tandem [6], and three genes encoding ubiquitin as a monomeric ubiquitin unit fused to ribosomal proteins: RPL40A (UBI1); RPL40B (UBI2); and RPS31 (UBI3) [7].

Furthermore, according to your suggestion, ubiquitin expressing genes in Metarhizium robertsii were also investigated by using the 76 aa protein sequence of yeast polyubiquitin protein (UBI4, YLL039C) to BLASTP against the M. robertsii genome database [18, 42], and the results show that there are three ubiquitin motif-containing genes in M. robertsii, that is, MrUBI1 (MAA_11504, monomeric ubiquitin fusion gene), MrUBI3 (MAA_04702, monomeric ubiquitin fusion gene), and MrUBI4 (MAA_02160, the polyubiquitin gene, 4 monomeric ubiquitin in tandem).

Taken together, the polyubiquitin gene (MrUBI4, MAA_02160) is a single copy in M. robertsii, and is focused in this study.

References

6.         Finley, D.; Ozkaynak, E.; Varshavsky, A. The yeast polyubiquitin gene is essential for resistance to high temperatures, starvation, and other stresses. Cell 1987, 48, 1035-1046, doi:10.1016/0092-8674(87)90711-2.

7.         Ozkaynak, E.; Finley, D.; Solomon, M.J.; Varshavsky, A. The yeast ubiquitin genes: a family of natural gene fusions. EMBO J. 1987, 6, 1429-1439.

8.         Gemayel, R.; Yang, Y.; Dzialo, M.C.; Kominek, J.; Vowinckel, J.; Saels, V.; Van Huffel, L.; van der Zande, E.; Ralser, M.; Steensels, J., et al. Variable repeats in the eukaryotic polyubiquitin gene ubi4 modulate proteostasis and stress survival. Nat. Commun. 2017, 8, 397, doi:10.1038/s41467-017-00533-4.

18.       Chen, Q.; Li, Y.B.; Wang, J.Z.; Li, R.; Chen, B.S. cpubi4 is essential for development and virulence in chestnut blight fungus. Front. Microbiol. 2018, 9, 1286, doi:10.3389/fmicb.2018.01286.

42.       Gao, Q.; Jin, K.; Ying, S.H.; Zhang, Y.J.; Xiao, G.H.; Shang, Y.F.; Duan, Z.B.; Hu, X.A.; Xie, X.Q.; Zhou, G., et al. Genome sequencing and comparative transcriptomics of the model entomopathogenic fungi Metarhizium anisopliae and M. acridum. PLoS Genet. 2011, 7, e1001264, doi:10.1371/journal.pgen.1001264.

(2)   Although genomic DNA has been examined for gene deletion, western blotting would be necessary to confirm the loss of ubiquitin expression, given to multiple copies of ubiquitin are present.

Response:  

Thank you for your suggestion. However, the polyubiquitin gene (MrUBI4, MAA_02160) is a single copy in M. robertsii.

Moreover, for gene deletion, positive transformants were usually confirmed via genomic PCR, RT-PCR, and Southern blot analysis [REF 1-3, 4], which are also performed in this study.

References

REF 4.  Islam, K.T.; Bond, J.P.; Fakhoury, A.M. FvSTR1, a striatin orthologue in Fusarium virguliforme, is required for asexual development and virulence. Appl. Microbiol. Biotechnol. 2017, 101, 6431-6445, doi:10.1007/s00253-017-8387-1.

REF 1.  Du, Y.; Jin, K.; Xia, Y. Involvement of MaSom1, a downstream transcriptional factor of cAMP/PKA pathway, in conidial yield, stress tolerances, and virulence in Metarhizium acridum. Appl. Microbiol. Biotechnol. 2018, 102, 5611-5623, doi:10.1007/s00253-018-9020-7.

REF 2.  Xie, M.; Xia, Y.; Cao, Y. The Rab GTPase activating protein Gyp2 contributes to UV stress tolerance in Metarhizium acridum. World J. Microbiol. Biotechnol. 2018, 34, 78, doi:10.1007/s11274-018-2457-0.

REF 3.   Wang, Z.; Jiang, Y.; Li, Y.; Feng, J.; Huang, B. MrArk1, an actin-regulating kinase gene, is required for endocytosis and involved in sustaining conidiation capacity and virulence in Metarhizium robertsiiAppl. Microbiol. Biotechnol.  2019, DOI: 10.1007/s00253-019-09836-6.

(3)   Ubiquitin expression levels in Comp strains to WT strain would need to be examined.

Response:  

Thank you for your suggestion.

In our study, the gene expression of the polyubiquitin gene MrUBI4 in WT, MT and Comp strains was examined by semi-quantitative RT-PCR analysis (see details in Figure S1C), and the results show that the MrUBI4 transcripts are completely undetectable in the MT strains, whereas it is detectable in the WT and Comp strains.

Minor concerns:

1.     Fig 2A: scale bars are needed.

Response:  

We apologize for this oversight. The corresponding scale bars were added in the revised Figure 2A.

2.     Fig 2C: statistical analyses are needed.

Response:  

Thank you for your suggestion.

Statistical analysis was conducted, and the corresponding sentence “The growth rates of the ∆MrUBI4 strain reduced significantly (Tukey’s test, p < 0.01) after 4 days post-inoculation.” was added to the legend of Figure 2C.

3.     Fig 4C: to understand why ubiquitin deletion affects UV sensitivity, DNA damage response genes would need to be examined.

Response:  

Thank you for your suggestion. In fact, the expression level of several DNA damage response genes had been analyzed to understand the effect on UV sensitivity according to the recent report [REF 1-2]. However, it is relatively small for the variation of expression level of DNA damage-related genes, compared to heat shock-related genes. Therefore, the data were not added to the Figure 4C at that time.

And now, the corresponding data were added to the revised Figure 4C.

References

REF 1.  Du, Y.; Jin, K.; Xia, Y. Involvement of MaSom1, a downstream transcriptional factor of cAMP/PKA pathway, in conidial yield, stress tolerances, and virulence in Metarhizium acridum. Appl. Microbiol. Biotechnol. 2018, 102, 5611-5623, doi:10.1007/s00253-018-9020-7.

REF 2.  Xie, M.; Xia, Y.; Cao, Y. The Rab GTPase activating protein Gyp2 contributes to UV stress tolerance in Metarhizium acridum. World J. Microbiol. Biotechnol. 2018, 34, 78, doi:10.1007/s11274-018-2457-0.

4.     Fig 5A: statistical significance is needed.

Response:

Thank you for your suggestion.

Statistical analysis was performed, and the corresponding sentence “The growth rates of the ∆MrUBI4 strain decreased significantly (Tukey’s test, p < 0.01) after 4 days post-inoculation.” was added to the legend of Figure 5A.

5.     The reviewer suggests the authors to seek help from English Professionals to eliminate typos and grammar mistakes in the manuscript.

Response:

Thank you for your suggestion. Previously, we have revised the whole manuscript using the Elsevier Language Editing Services (Express).

We have taken your comment into consideration, and we have further revised carefully the whole manuscript to avoid typos and grammar mistakes in the revised manuscript, such as L90-92, L100, L104, L115-116, L127, L213, L260-267, and L362-373.

Round 2

Reviewer 1 Report

After revision, I believe that the authors have adequately addressed most of the concerns. Therefore, I would recommend this work to publish in Genes.